# Swiprosin-1/EFhD-2 Expression in Cardiac Remodeling and Post-Infarct Repair: Effect of Ischemic Conditioning

**DOI:** 10.3390/ijms21093359

**Published:** 2020-05-09

**Authors:** Zoltán Giricz, András Makkos, Rolf Schreckenberg, Jochen Pöling, Holger Lörchner, Krisztina Kiss, Péter Bencsik, Thomas Braun, Rainer Schulz, Péter Ferdinandy, Klaus-Dieter Schlüter

**Affiliations:** 1Cardiometabolic Research Group and MTA-SE System Pharmacology Research Group Department of Pharmacology and Pharmacotherapy, Semmelweis University, 1085 Budapest, Hungary; zoltan.giricz@pharmahungary.com (Z.G.); andras.makkos@pharmahungary.com (A.M.); peter.ferdinandy@pharmahungary.com (P.F.); 2Pharmahungary Group, H-6720 Szeged, Hungary; peter.bencsik@pharmahungary.com; 3Department of Physiology, Justus-Liebig-University, 35390 Giessen, Germany; rolf.schreckenberg@physiologie.med.uni-giessen.de (R.S.); rainer.schulz@physiologie.med.uni-giessen.de (R.S.); 4Department of Cardiac Development and Remodeling, Max Planck Institute for Heart and Lung Research, Ludwigstrasse 43, 61231 Bad Nauheim, Germany; Jochen.Poeling@mpi-bn.mpg.de (J.P.); Holger.Loerchner@mpi-bn.mpg.de (H.L.); Thomas.Braun@mpi-bn.mpg.de (T.B.); 5Department of Pharmacology and Pharmacotherapy, University of Szeged, H-6721 Szeged, Hungary; krisztina.kiss@pharmahungary.com

**Keywords:** cardiac regeneration, cardiac protection, miR-34

## Abstract

Swiprosin-1 (EFhD2) is a molecule that triggers structural adaptation of isolated adult rat cardiomyocytes to cell culture conditions by initiating a process known as cell spreading. This process mimics central aspects of cardiac remodeling, as it occurs subsequent to myocardial infarction. However, expression of swiprosin-1 in cardiac tissue and its regulation in vivo has not yet been addressed. The expression of swiprosin-1 was analyzed in mice, rat, and pig hearts undergoing myocardial infarction or ischemia/reperfusion with or without cardiac protection by ischemic pre- and postconditioning. In mouse hearts, swiprosin-1 protein expression was increased after 4 and 7 days in myocardial infarct areas specifically in cardiomyocytes as verified by immunoblotting and histology. In rat hearts, swiprosin-1 mRNA expression was induced within 7 days after ischemia/reperfusion but this induction was abrogated by conditioning. As in cultured cardiomyocytes, the expression of swiprosin-1 was associated with a coinduction of arrestin-2, suggesting a common mechanism of regulation. Rno-miR-32-3p and rno-miR-34c-3p were associated with the regulation pattern of both molecules. Moreover, induction of swiprosin-1 and ssc-miR-34c was also confirmed in the infarct zone of pigs. In summary, our data show that up-regulation of swiprosin-1 appears in the postischemic heart during cardiac remodeling and repair in different species.

## 1. Introduction

Swiprosin-1, also known as EF-Hand domain family member D2 (EFhd2), acts as a calcium sensor that stabilizes F-actin filaments by blocking binding sites of cofilin [1]. Cofilin is needed for the depolymerization of F-actin [2]. Expression of swiprosin-1 has already been identified in lymphocytes, B-cells, mast cells, T-cells, neurons, podocytes, and cardiomyocytes [3,4,5,6,7,8,9]. Furthermore, a drosophila-specific homologue of human swiprosin-1 participates in skeletal muscle formation [10]. In immune cells, it triggers the formation of lamellopodia, which is required for migration of macrophages [11]. In neurons, swiprosin-1 regulates the cargo capacity of kinesin and participates in neuronal outgrow [1]. In cardiomyocytes, swiprosin-1 is required for the formation of pseudopodia-like structures [9,12]. Pseudopodia-like structures enable adult rat ventricular cardiomyocytes to adapt to culture dishes, to stay alive for days, and to rearrange their sarcomeres by adaptation from the three-dimensional heart to the two-dimensional surrounding of culture dishes. At the level of swiprosin-1 regulation, changes in the mRNA levels went parallel with those of protein expression, suggesting transcriptional regulation. However, the more important consequence of this observation is that this process mimics aspects of cardiac remodeling. Extensive cardiac remodeling with global cardiac changing in gene expression occurs subsequently to myocardial infarction (MI) when repair processes occur [13,14,15]. Therefore, in the present study we investigated the expression of swiprosin-1 in cardiac tissue during subsequent cardiac remodeling.

Among the common features between cardiac tissue repair and cellular adaptation of terminal differentiated cardiomyocytes to culture conditions are a desensitization of β-adrenoceptors that can be modified by ischemic postconditioning (IPoC) [16]. Similarly, β-adrenoceptor responsiveness is reduced during cellular adaptation of cardiomyocytes to cultivation. In line with this observation, swiprosin-1 was found to be strongly coregulated under culture conditions with molecules that regulate β-adrenoceptor responsiveness, such as β-arrestin-2 [9]. This coregulation suggests a common mechanism of regulation for both genes. Nowadays, micro-RNAs (miRs) have been identified as potential molecules that coordinate the several proteins acting together to modify the phenotype of individual cells. They bind to mRNA and abrogate subsequent translation into protein. Therefore, we also screened for potential miRs that may commonly be involved in the coregulation of swiprosin-1 and β-arrestin-2.

## 2. Results

### 2.1. Swiprosin-1 Expression In Post-Infarcted Mice

Although swiprosin-1 expression and its role in cellular adaptation has been documented before [9], the relevance of these in vitro findings for cardiac repair processes remained unclear. Therefore, we first investigated the expression of swiprosin-1 in post-MI hearts. Experiments were performed on mouse hearts previously used for investigating the role of oncostatin M in cardiac repair, because in cultured cardiomyocytes swiprosin-1 and oncostatin M are coregulated. Immunoblotting of myocardial tissue samples of infarcted mice specified an increased expression of swiprosin-1 within the infarcted region when compared with the noninfarcted remote myocardium (Figure 1A). Of note, elevated levels of swiprosin-1 were maintained throughout the first week of cardiac remodeling (Figure 1A). Immunofluorescence analysis of infarcted hearts further allocated increased expression of swiprosin-1 in preserved cardiomyocytes surrounding the infarct scar, whereas swiprosin-1 was barely detectable in cardiomyocytes of distal noninfarcted regions (Figure 1B).

### 2.2. Induction of Swiprosin-1 in Rat Hearts after Ischemia/Reperfusion

In order to address the question of whether induction of cardiac expression of swiprosin-1 in post ischemic hearts is related to the extent of cardiac repair, swiprosin-1 mRNA expression was analyzed next in rat hearts 7 days after ischemia/reperfusion (I/R). These hearts have previously been used to establish the effect of ischemic pre- and postconditioning on cardiac remodeling [17]. Here, we show that swiprosin-1 mRNA expression is induced by I/R. However, either ischemic preconditioning (IPC) or IPoC abrogate this induction (Figure 2A). IPC and IPoC have proven effects on infarct sizes as quantified by troponin I plasma levels 1 h after reperfusion [17]. The mRNA expression of swiprosin-1 showed a linear relationship to the infarct size, suggesting that the amount of expression is related to cardiac repair processes (Figure 2B). In vitro, swiprosin-1 mRNA expression in cardiomyocytes is associated with a similar expression level of arrestin-2. In the cardiac tissue used to quantify the mRNA expression of swiprosin-1, a similar linear correlation between swiprosin-1 and arrestin-2 was found (Figure 2C).

Coregulation of swiprosin-1 and arrestin-2 suggests a common mechanism of regulation, such as by miR. In the rat heart tissue used for this study, 364 miRs were constitutively detected in all samples. Among them, 27 miRs were either positively (*n* = 15) or negatively (*n* = 12) associated with the mRNA expression of swiprosin-1 (Table 1). However, only 12 of 27 miRs showed an ischemia/reperfusion-dependent regulation such as swiprosin-1, and only in two cases could the induction of miR be abrogated by IPC or IPoC as found for swiprosin-1. These two most likely candidates were rno-miR-32-3p and rno-miR-34c-3p (Figure 3A,B).

As expected from the observed coregulation, the mRNA expression of swiprosin-1 showed a linear correlation with the amount of rno-miR-32-3p and rno-miR-34c-3p in these samples (Figure 4A). Moreover, both miRs showed a linear correlation with the infarct size (quantified as troponin I plasma concentration 1 h after I/R) and with arrestin-2 mRNA expression as well (Figure 4B,C).

In the cardiac tissue used for this analysis, mRNA expression of 72 different genes related to cardiac biology was analyzed (see ref. 17). A detailed analysis of further coregulated mRNAs with both miRs was performed by investigating a potential linear correlation between mRNAs and miRs. Interestingly, rno-miR-32-3p showed a positive correlation with genes associated with hypertrophy and fibrosis and a negative correlation with sodium–calcium exchanger (NCX) (Table 2). The data suggest that the increased expression of rno-miR-32-3p contributes to the maladaptive phenotype of cardiac adaptation in these hearts. In contrast, rno-miR-34c-3p displayed a nearly exclusive coregulation with swiprosin-1 and arrestin-2 (Table 2).

### 2.3. Regulation of Swiprosin-1 Expression in Pig Hearts

The aforementioned experiments have shown that swiprosin-1 is up-regulated during cardiac repair processes in small rodents. To investigate the relevance for other species as well, the investigation was extended by analysis of swiprosin-1 expression in pig hearts. This analysis was performed on pig heart samples previously analyzed in more detail [18,19]. Tissue samples were taken 3 h and 3 days after reperfusion from the border zone, infarct zone, and unaffected control zone. Again, swiprosin-1 mRNA expression was induced at the late time-point (day 3) in the infarct zone (Figure 5A). In comparison, expression of proinflammatory markers such as MCP-1 was induced already after 3 h (Figure 5B). Finally, in the reperfused heart, ssc-miR-34c was again increased by ischemia/reperfusion, and this was again abrogated by IPC and IPoC.

## 3. Discussion

Swiprosin-1 is expressed in various tissues, but although its expression and function in adult rat ventricular cardiomyocytes have been reported before, a proof of concept is still lacking that will show that swiprosin-1 is expressed in cardiac tissue and furthermore differentially regulated during active repair processes. Postinfarct remodeling requires extensive cardiac reconstruction and is therefore suitable to address this question. Using different conditioning protocols (IPC and IPoC), infarct sizes and thereby the amount of repair process can be modified. In line with these assumptions, the main findings of the current study are that swiprosin-1 expression is induced during cardiac repair processes in mice, rats, and pigs. Moreover, the expression of swiprosin-1 is strongest in those areas of the heart where cardiac remodeling occurs, that is, in the border zone of MI mice hearts and in the infarct zone of reperfused rat and pig hearts. Finally, histology identified cardiomyocytes as the main side of expression.

Our study is based on the assumption that swiprosin expression in cardiomyocytes is transcriptionally regulated. This argument comes from our previous study showing that targeting swiprosin mRNA by siRNA reduces swiprosin protein expression as well. In our current study, we show swiprosin protein expression in mice and confirmed its expression on mRNA in rat and pig. This expression is associated with miR expression, indicating a complex regulatory network in cardiomycytes. Nevertheless, on the basis of the material used in this study we could not directly confirm a strict coregulation of swiprosin mRNA and swiprosin protein for the three species, and this is of course a limitation of the study.

In the process of cardiac remodeling, a coordinative regulation of genes that encode for proteins required for cardiac contractility and excitation is required. As a potential mechanism to coordinate such expression, miRs are interesting molecules as the predicted consensus sequence of these molecules potentially targets multiple mRNAs. It was already known that in cardiomyocytes swiprosin-1 is coregulated with the mRNA expression of proteins linked to β-adrenoceptor desensitization. A desensitization of β-adrenoceptors can occur under conditions of cardiac cell cultivation and in the postischemic heart [16]. Nevertheless, coregulation of cardiac expression of swiprosin-1 and arrestin-2 in vivo has not been demonstrated before. Screening potential miRs that may trigger this coregulation, two miRs were identified. Both showed a similar type of regulation of expression to swiprosin-1, namely, rno-miR-32-3p and rno-miR-34c-3p. The amount of expression of both miRs correlates with swiprosin-1 and arrestin-2.

Rno-miR-32-3p was also associated with other genes that are well known to affect cardiac remodeling. The level of miR expression correlated with genes involved in fibrosis (biglycan, collagen-1, collagen-3, and fibronectin), inflammation (iNOS, arginase-2), and receptor activity (RAMP-3). Biglycan and fibronectin can act as a modulator of TGF-β1 activity, and in line with this function the expression of collagen increased [20,21]. RAMPs, but not RAMP-3, are differentially expressed in cardiomyocytes under hypertensive conditions [22]. On the other hand, increased expression of rno-miR-32-3p was inversely correlated with the expression of sarcomeric proteins (MHC-α, troponin T), transporters of sodium (NCX, Scl5a3), and that of fatty acid metabolism (HADHA, UCP-2). While HADHA is required for fatty acid metabolism, down-regulation of UCP-2 will favor glucose utilization [23]. Therefore, the down-regulation of HADHA and UCP-2 both initiate a metabolic switch into the direction of glucose metabolism. In general, the common regulation of swiprosin-1 with other proteins known to be differentially regulated during cardiac remodeling suggests that swiprosin-1 is part of this program. However, as most of the adaptations (fibrosis, metabolic switch, and sarcomere remodeling) are associated with maladaptive function, the overall function of swiprosin-1 may be maladaptive as well. As swiprosin-1 participates in cardiomyocyte dedifferentiation, a mechanism that leads to impaired cardiac function when occurring in a chronic rather than acute way, this may indeed be the case. In this way, the lower expression of swiprosin-1 after IPC and IPoC will be the consequence of smaller infarct sizes.

In contrast, rno-miR-34c-3p was not associated with maladaptive gene reprogramming such as rno-miR-32p. This miR, in addition to swiprosin-1, was associated with down-regulation of four different ion transporters. Of note, I/R-dependent regulation of miR-34c was found in rat and pig tissues. Finally, both miRs identified in this study have not yet been associated with myocardial remodeling and represent another new finding of this study.

The design of the current study does not allow identifying potential mechanisms by which swiprosin-1 may affect the remodeling process. However, it has been shown before that dedifferentiation processes are needed for regeneration of cardiomyocytes. Such processes depend on oncostatin M as well as of swiprosin-1 [24]. In this context, it was also shown that swiprosin-1 and oncostatin M are coexpressed and coregulated in cardiomyocytes in vitro [12]. Oncostatin M may link the initial inflammatory process during cardiac repair to cardiac remodeling, and this process may require swiprosin-1 to arrange sarcomeric structures.

In the current study, swiprosin expression was found in the same region of mouse tissue and at the same time-point as oncostatin M (this study and ref. 24). Oncostatin M is released by immune cells like macrophages and triggers the release of Reg3β from cardiomyocytes. Reg3β directs more oncostatin M, releasing immune cells to this area. At the same time, oncostatin M triggers the dedifferentiation of cardiomyocytes. This process requires swiprosin [9]. Therefore, oncostatin M-producing immune cells like macrophages may potentiate the induction of swiprosin expression at the site of cardiac differentiation.

In conclusion, this study highlights the tissue-specific expression of swiprosin-1 in postinfarct hearts, and the way of regulation suggests an important role in cardiac repair.

## 4. Materials and Methods

### 4.1. Animal Experiments

All types of analysis in this study were performed on tissue samples from previous experiments. Therefore, no new animal experiments were performed for this study. The samples for analysis of mice tissue were generated in a mouse MI model as described before [24]. Briefly, the model used a permanent ligature of the LAD in mice with a subsequent 4 and 7 days follow-up before the mice were sacrificed. All relevant functional data were published before [24]. The samples for analysis of rat tissue were generated in a rat model of ischemia/reperfusion as described [16]. This model uses a 30 min LAD occlusion with subsequent reperfusion for 7 days. All relevant functional data were published before [16,17]. Infarct sizes were quantified by increased plasma troponin I levels 60 min after reperfusion as described before [16]. The samples for analysis of pig tissue were generated in a pig model of I/R as described [18,19]. This model is based on a 90 min percutaneous balloon occlusion of the left anterior descending coronary artery, followed by balloon deflation in anaesthetized pigs and subsequent reperfusion for 3 h or 3 d.

### 4.2. Western Blot and Histology on Mouse Samples

Infarcted hearts of adult mice were harvested and dissected in an infarction zone and remote myocardium (basal part of the interventricular septum) for Western Blot analysis. Fractionated samples were sonicated in lysis buffer (0.1 M Tris-HCl pH 8.8, 0.01 M EDTA, 0.04 M DTT, 10 % SDS, and pH 8.0) supplemented with protease inhibitors. Twenty micrograms of each protein sample were loaded on Gradient NuPAGE 4-12% Bis-Tris gels (Invitrogen) to perform SDS-PAGE. After transferring onto nitrocellulose membranes (Invitrogen), immunoreactive proteins were visualized on the ChemiDocXRS+ System (BioRad) and analyzed by using the Image Lab software (BioRad). The following antibodies for Western Blot analysis were used: goat anti-Swiprosin-1 antibody (Sigma-Aldrich, catalogue no. SAB2501007) and rabbit anti-Pan-actin antibody (Cell Signaling Technology, catalogue no. 4968). Secondary antibodies conjugated with horseradish peroxidase were purchased from R&D Systems.

The method for immunohistochemical analysis of murine infarcted hearts has been previously described (24). The following antibodies for immunofluorescence staining were used: goat anti-Swiprosin-1 antibody (Sigma-Aldrich, catalogue no. SAB2501007) and rabbit anti-α-sarcomeric-actinin monoclonal antibody (Clone: EA-52, Sigma-Aldrich, catalogue no. A7732). For nuclear counterstain, 4′,6-diamidino-2-phenylindole (DAPI) was used (Sigma Aldrich, catalogue no. D9542). Lectin from Ulex europaeus (L4895; catalogue no. L4895; Sigma Aldrich) was used to detect intact cells.

### 4.3. RT-PCR on Rat and Pig Tissue

The method on RNA isolation and real-time reverse-transcriptase polymerase chain reaction (RT-PCR) used to quantify the mRNA expression has been described before in detail [16]. A list of all rat primers used in the study has also published before [17] Table 3.

### 4.4. Statistics

Group comparison was performed by One-Way-ANOVA and Student–Newman–Keuls post hoc analysis. If the probability to belong to equal groups is below *p*≤0.05, this is indicated by different letters in the figures. Correlation analysis was performed by Pearson correlation. Exact *p* values for correlation analysis are given in the figures to the legend.

## Figures and Tables

**Figure 1 ijms-21-03359-f001:**
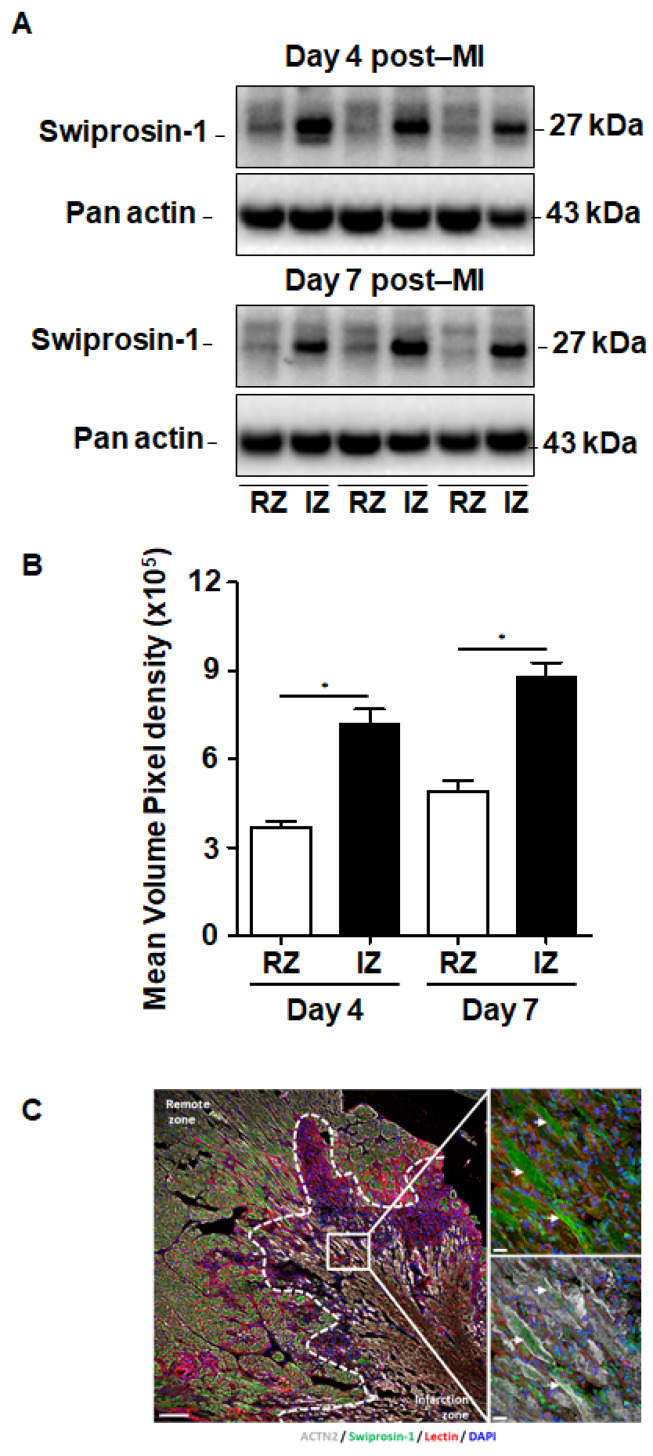
Expression of swiprosin-1 after the onset of myocardial infarction in adult mice. (**A**) Representative Western Blot analysis of swiprosin-1 in myocardial tissue from adult mice fractionated in noninfarct remote zone (RZ) and infarction zone (IZ) at days 4 and 7 after myocardial infarction (post-MI). (**B**) Quantification of the two Western Blots showed by A with *, *p* < 0.05 vs. RZ; (**C**) Immunofluorescence analysis of swiprosin-1 (green) in infarcted hearts from mice 4 days post-MI. Viable cardiomyocytes were stained with ACTN2 (grey) and lectin (red). 4′,6-diamidino-2-phenylindole (DAPI) was used for nuclear staining (blue). White arrows classify swiprosin-1 immunopositive cardiomyocytes. Scale bars: 100 µm and 20 µm in magnified sections.

**Figure 2 ijms-21-03359-f002:**
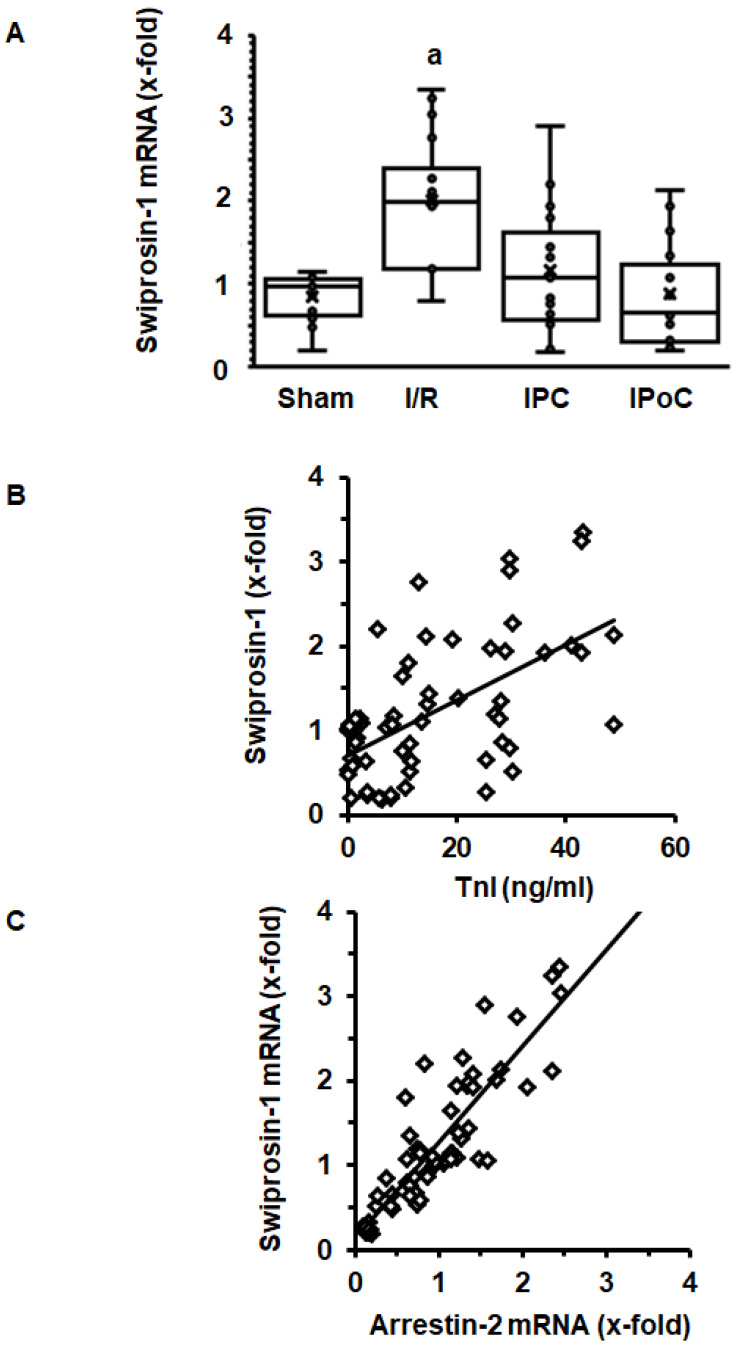
Expression of swiprosin-1 mRNA in the left ventricle from rats seven days after I/R. (**A**), Box plots showing the level of expression and distribution of samples from sham surgery (Sham), ischemia/reperfusion (I/R), ischemic preconditioning (IPC), and ischemic postconditioning (IPoC). Data are analyzed by One-Way-ANOVA with Student–Newman–Keuls post-hoc analysis (a; *p* < 0.05 vs. all other groups). (**B**); Correlation between infarct sizes as quantified by troponin I release into the plasma 60 min after reperfusion and swiprosin-1 mRNA expression. (**C**) Correlation between arrestin-2 mRNA expression and swiprosin-1 mRNA expression.

**Figure 3 ijms-21-03359-f003:**
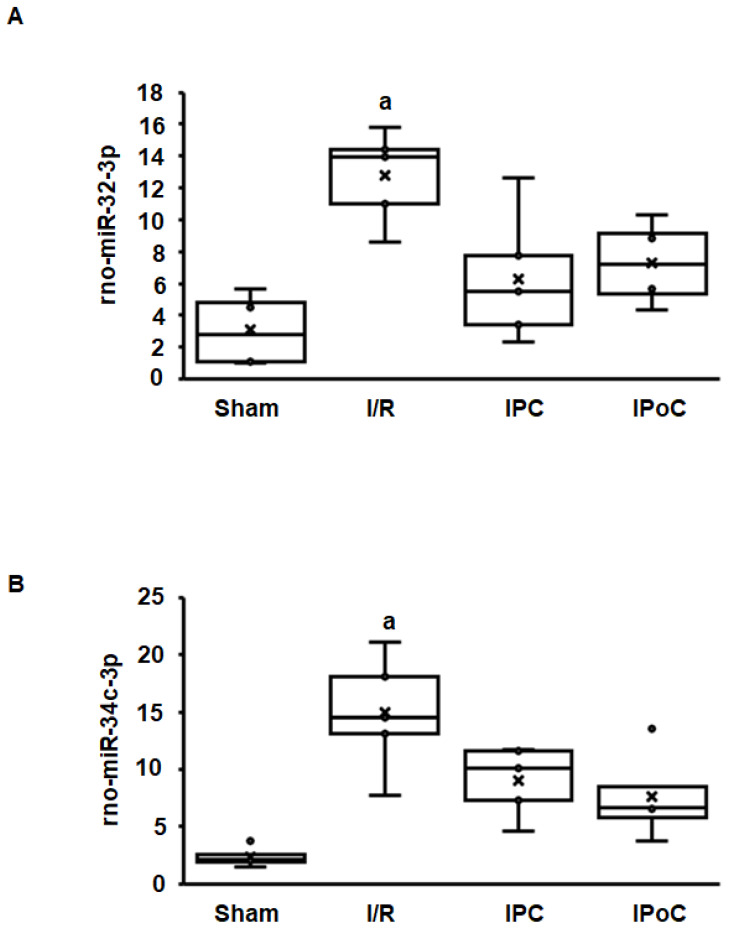
Expression of rno-miR-32-3p (**A**) and rno-miR-34c-3p (**B**) in the left ventricle from rats seven days after I/R. A). Box plots showing the level of expression and distribution of samples from sham surgery (Sham), ischemia/reperfusion (I/R), ischemic preconditioning (IPC), and ischemic postconditioning (IPoC). Data are analyzed by One-Way-ANOVA with Student–Newman–Keuls posthoc analysis (a; *p* < 0.05 vs. all other groups).

**Figure 4 ijms-21-03359-f004:**
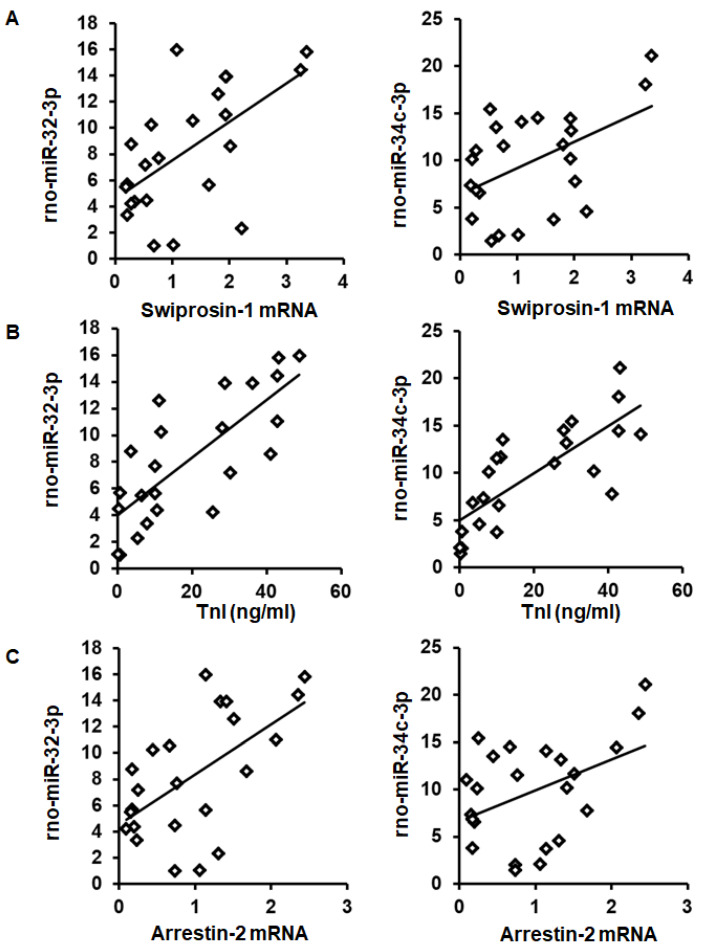
Correlation between swiprosin-1 mRNA (**A**), infarct size (as quantified by troponin I plasma concentration) (**B**), and arrestin-2 (**C**), with the abundancy of rno-miR-32-3p and rno-miR-34c-3p.

**Figure 5 ijms-21-03359-f005:**
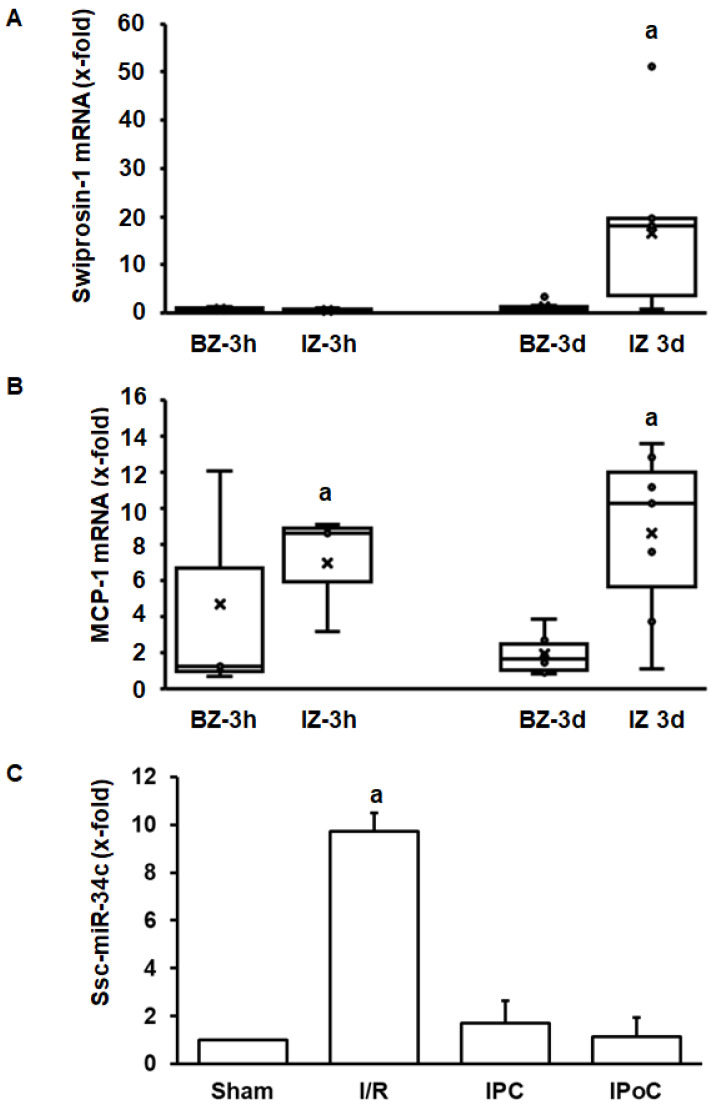
Expression of swiprosin-1, MCP-1, and ssc-miR-34c in the border zone (BZ) or infarct zone (IZ) three hours or 3 days after reperfusion. The quantification for ssc-miR-34c was performed in the infarct zone 3 h after reperfusion.

**Table 1 ijms-21-03359-t001:** Correlation between swiprosin-1 mRNA and micro-RNAs (miRs).

rno-miR	R-Coefficient	*p*-Value	I/R	IPC/IPoC
Positive Correlation				
32-3p	*r* = 0.596	*p* = 0.003	yes	yes
34c-3p	*r* = 0.495	*p* = 0.016	yes	yes
18a-3p	*r* = 0.465	*p* = 0.025	yes	no
23b-5p	*r* = 0.475	*p* = 0.022	yes	no
27b-5p	*r* = 0.436	*p* = 0.038	yes	no
100-3p	*r* = 0.431	*p* = 0.040	no	-
146a-5p	*r* = 0.454	*p* = 0.029	no	-
296-5p	*r* = 0.474	*p* = 0.022	no	-
324-5p	*r* = 0.520	*p* = 0.011	no	-
342-3p	*r* = 0.516	*p* = 0.012	no	-
455-3p	*r* = 0.453	*p* = 0.030	no	-
497-5p	*r* = 0.458	*p* = 0.028	no	-
505-5p	*r* = 0.436	*p* = 0.038	no	-
532-3p	*r* = 0.514	*p* = 0.012	no	-
674-5p	*r* = 0.616	*p* = 0.002	no	-
Negative Correlation				
7b				
24-2-5p	*r* = -0.475	*p* = 0.022	no	-
29b-3p	*r* = -0.425	*p* = 0.043	yes	no
33-5p	*r* = -0.563	*p* = 0.005	yes	no
99a-5p	*r* = -0.431	*p* = 0.040	yes	no
145-3p	*r* = -0.520	*p* = 0.011	yes	no
150-5p	*r* = -0.414	*p* = 0.050	yes	no
186-5p	*r* = -0.425	*p* = 0.043	no	-
194-5p	*r* = -0.495	*p* = 0.016	yes	no
339-3p	*r* = -0.447	*p* = 0.033	yes	no
345-3p	*r* = -0.426	*p* = 0.042	no	-
3068-5p	*r* = -0.422	*p* = 0.045	yes	no

**Table 2 ijms-21-03359-t002:** Correlation between miRs and mRNAs.

**rno-miR-32-3p**		
Swiprosin-1	*r* = 0.596	*p* = 0.003
Arrestin-2	*r* = 0.585	*p* = 0.003
Biglycan		
Collagen-1	*r* = 0.581	*p* = 0.004
Collagen-3	*r* = 0.515	*p* = 0.020
Fibronectin	*r* = 0.452	*p* = 0.030
Arginase-2	*r* = 0.434	*p* = 0.038
iNOS		
RAMP-3		
MHC-α		
NCX		
Troponin T	*r*= -0.470	*p* = 0.023
HADHA		
UCP-2		
Slc5a3		
**rno-miR-34c-3p**		
Swiprosin-1	*r* = 0.495	*p* = 0.016
Arrestin-2	*r* = 0.440	*p* = 0.036
ANP		
MHC-α		
Elfn2		
Kcnh5		
Slc5a3		
Socs7		

Abbreviations: NOS, nitric oxide synthase; RAMP, receptor associated modifying protein; ANP, atrial natriuretic peptide; MHC; myosin heavy chain; Elfn, extracellular leucine rich repeat and fibronectin type III domain containing; Kcnh, potassium voltage-gated channel, subfamily H; Slc, solute carrier; Socs, suppressor of cytokine signaling.

**Table 3 ijms-21-03359-t003:** List of porcine primers used in this study.

Beta-2-Mircoglobulin	forward: CGT GGC CTT GGT CCT GCT CG	reverse: TCC GTT TTC CGC TGG TGT GC
EFhD2	forward: TCC GGG AGT TCC TCC TGA TT	reverse: AAG CTC TTC GCT CCC TTG AC
MCP-1	forward: CAG CCA CCT TVT GCA CCC AGG	Reverse: CAC AGA TCT CCT TGCCCG CGA

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
