# Peer review of "Swiprosin-1/EFhD-2 Expression in Cardiac Remodeling and Post-Infarct Repair: Effect of Ischemic Conditioning"

_ijms, 2020, doi:10.3390/ijms21093359_

Round 1

Reviewer 1 Report

The topic of the manuscript by Giricz et al. is focused on cardiac remodeling after ischemic conditions. From the point of view that myocardial infarction remains one of the leading causes of death submitted paper could be considered as novel and important. On the other hand data are already partially published. I have some concerns and comments:

  • I recommend to take out all information already published from results, could be use in discussion
  • It is not standard to use 3 different animal model in which different methodology were used and compared (immunobloting and histology from mouse model vs. RT-PCR from rat and pig model). I request to add the rest (immunobloting and histology from rat and pig and RT-PCR from mice)
  • What is the ratio of increased expression of swiprosin-1 in mice heart?

Author Response

You wrote: The topic of the manuscript by Giricz et al. is focused on cardiac remodeling after ischemic conditions. From the point of view that myocardial infarction remains one of the leading causes of death submitted paper could be considered as novel and important. On the other hand data are already partially published.

Response: We thank the reviewer for the evaluation of our manuscript. Please note that none of the results of the current study have been published before. However, the reviewer is right in that we used tissue samples from previous experimental studies to analyze changes in swiprosin expression. Please see our detailed point-by-point comments for clarification.

You wrote: I recommend to take out all information already published from results, could be use in discussion

Response: The reviewer has obviously the impression that the manuscript contains data that have already been published; however this is not the case. Nevertheless, the reviewer is right in that we used tissue samples from previous experiments (which have been performed to test different hypotheses) and as such the models used have already been undergone extensive reviews.

The mouse tissue used (chapter 2.1.) were generated to characterize the role of oncostatin M (OSM) in cardiac remodeling (see ref. 24 and Lörchner et al., 2018; Int J Cardiol 258: 7-13). From tissue slices that were not used in the previous studies and from stored protein samples not used before, we now performed histology and Western blots addressing the role of swiprosin. We used this material for the following reason: Swiprosin has been shown by us to be involved in cardomyocytes dedifferentiation (ref. 9). OSM is a cytokine expressed in the border zone where cardiac remodeling is active. OSM triggers dedifferentiation of cardiomyocytes. Therefore, we now tested whether or not swiprosin is co-activated in this area at the same time where OSM is induced.

The rat tissue (chapter 2.2.) was previously used to characterize the effect of ischemic pre- and post-conditioning (ref. 16). Next we analyzed the expression of miRs (ref. 17). In the present study we now compared and associated the expression of miRs and swiprosin.

The pig tissue (chapter 2.3.) was used to monitor miR expression in post-myocardial remodeling in a different species to verify that the results obtained in chapter 2.2. are not limited to rats. Although data on infarct size and troponin I release have been previously published, we now utilize the information only to show correlation with new parameters measured in the present study. Since the fact of reuse of previous generated material is stated multiple times in the text, we believe that it is not against ethical norms to use them to prove novel findings based on samples obtained from a previous study. This is a usual practice when in vivo animal experiments or clinical studies are further evaluated after publication of study design or basic phenotype of an experimental model. Moreover, this kind of interpretation of novel findings is highly recommended and supported by ethical consideration of the 3R rule disclosed in the EU Directive (2010/63/EU).

You wrote: It is not standard to use 3 different animal model in which different methodology were used and compared (immunobloting and histology from mouse model vs. RT-PCR from rat and pig model). I request to add the rest (immunobloting and histology from rat and pig and RT-PCR from mice)

Response: It was important for us to show that similar regulation of swiprosin occurs in a model of myocardial infarction (permanent ligature) and in post-ischemic remodeling (reperfusion injury). Therefore, we used permanent ligature (mouse) and ischemia/reperfusion models (rat, pig). Furthermore, the current approach allows to prove whether the observed effects are species independent. Due to the design of the study we do not have tissue samples prepared for histology from rat and pig tissue. Similarly, we do not have RNA isolated from the mouse tissue that was either prepared for histology or for protein analysis. However, we consider this not as critical as we conformed co-expression of swiprosin in areas previously positive to OSM (a cytokine triggering dedifferentiation of cardiomyocyes). Furthermore, we know from our previous study that swiprosin is transcriptionally regulated (ref. 9).

However, we add a point (limitation of the study) that addresses this point. Please read on line 238: “Our study is based on the assumption that swiprosin expression in cardiomyocytes is transcriptionally regulated. This argument comes from our previous study showing that targeting swiprosin mRNA by siRNA reduces swiprosin protein expression as well. In our current study we show swiprosin protein expression in mice and confirmed its expression on mRNA in rat and pig. This expression is associated with miR expression indicating a complex regulatory network in cardiomycytes. Nevertheless, on the basis of the material used in this study we could not directly confirm a strict co-regulation of swiprosin mRNA and swiprosin protein for the three species and this is of course a limitation of the study.”

You wrote: What is the ratio of increased expression of swiprosin-1 in mice heart?

Response: We have add these data to figure 1.

Reviewer 2 Report

The paper by Giricz by et al examines the localisation of swiprosin-1 in ischemic cardiac tissue from material that was already available from earlier experiments analyzing cardiac remodeling. The material came from mice, rats and pigs. The results are interesting and highlight the repair process after myocardial infarction. The experiments were well performed and the paper is well written.

Minor
- did the authors stain for macrophages in the ischemic tissue? Is there a relation with the expression of swiprosin-1?
- I would like to suggest that the authors discuss the link between swiprosin-1, the found miRNAs and macrophages in the context

Author Response

You wrote: The paper by Giricz by et al examines the localisation of swiprosin-1 in ischemic cardiac tissue from material that was already available from earlier experiments analyzing cardiac remodeling. The material came from mice, rats and pigs. The results are interesting and highlight the repair process after myocardial infarction. The experiments were well performed and the paper is well written.

Response: We thank the reviewer for evaluation of our manuscript. Please see our detailed point-by-point comments for clarification of the minor points mentioned in your review.

You wrote: did the authors stain for macrophages in the ischemic tissue? Is there a relation with the expression of swiprosin-1?

Response: With regard to the first part of the study (chapter 2.1: mouse tissue) this was done previously (ref. 24) in the same tissue where oncostatin M was identified as a trigger of cardiomyocytes dedifferentiation that co-localizes also with swiprosin (this study). In case of rat and pig tissue we did not stain the tissue for macrophages. However, in the rat tissue MMP12, a macrophage-specific elastase, is co-regulated with swiprosin (mRNA level). This may indicate a similar participation of macrophages in this process.

You wrote: I would like to suggest that the authors discuss the link between swiprosin-1, the found miRNAs and macrophages in the context

Response: We thank you for your comment. We extended the discussion according to your suggestion. Please read on line 287: “In the current study swiprosin expression was found in the same region of mouse tissue and at the same time-point as oncostatin M (this study and ref. 24). Oncostatin M is released by immune cells like macrophages and triggers the release of Reg3β from cardiomyocytes. Reg3β directs more oncostatin M releasing immune cells to this area. At the same time oncostatin M triggers the dedifferentiation of cardiomyocytes. This process requires swiprosin (ref. 9). Therefore, oncostatin M producing immune cells like macrophages may potentiate the induction of swiprosin expression at the site of cardiac differentiation.”

Round 2

Reviewer 1 Report

Authors replied all points and I agree to publish the manuscript in currect form.